# Performance Factors in Dinghy Sailing: Laser Class

**DOI:** 10.3390/ijerph16244920

**Published:** 2019-12-05

**Authors:** Israel Caraballo, José Luis González-Montesinos, Antonio Alías

**Affiliations:** 1Department of Sport Sciences, University of Cádiz, 11519 Cádiz, Spain; jgmontesinos@uca.es; 2Department of Sport Sciences, University of Almería, 04120 Almería, Spain; aag344@ual.es

**Keywords:** performance, sport sailing, strength, resistance, dinghy hiking

## Abstract

Despite the relationship between performance and anthropometric characteristics, strength, and endurance in the action of dinghy hiking, there is no equation to predict the position obtained in competition. The purpose of this study was to examine the effects of anthropometric characteristics, strength, and endurance on the performance of the sailor. Twenty-nine male sailors of the Laser class were evaluated according to age, navigation experience, strength and resistance tests in a simulator, body weight, size, sitting height, Body Mass Index (BMI), body fat percentage, trochanteric length, thigh length, tibial length, foot length, abdominal perimeter, and upper thigh perimeter. The results show that the variables were related to performance are age, navigation experience, height, and length of the thigh. The variables that are most related to performance are age and sailing experience. Seventy-six percent of the performance can be estimated using the following equation: 311.971 + (−1.089 × height) + (−1946 × age) + (−1.537 × thigh length). Performance in the Laser class will be determined by the tactics (age and sailing experience) and the morphological characteristics of the sailor (height and sitting height).

## 1. Introduction

Dinghy sailing is an Olympic sport in which factors such as body composition, fitness, technique, and tactics determine performance [1]. Among the different classes of sailing sports, the Laser boat is in the monohull category and is manned by a single athlete. It was designed by Bruce Kirby in 1969, and it is the only authorised vessel in the European and international circuits, according to the International Sailing Federation (ISAF). In this type of boat, performance is significantly related to the ability of the sailor to overcome the forces that tend to destabilize the boat, which are generated by the action of the wind on the sails, producing a lateral inclination called heel [2]. To overcome these forces and to thus stabilize the boat, the sailor performs a specific technical gesture called hiking bench. With this gesture, the body is used as a lever arm, placing the feet in straps that are located in the center of the boat while leaning and taking the rest of the body out of the side in order to balance the moment of the resulting forces. The main objective of this maneuver is to maintain stability, which allows the sailor to use the force of the wind to increase the speed of the boat to its maximum during most of the route [3]. With winds greater than eight knots and in certain circumstances, the sailor can make this maneuver for 94% of the total sailing time [4].

Regarding the hiking bench, there are three main positions, which are defined according to the angulation between the hip and the trunk [5]. The first position is called sitting and comprises an angulation between 90° and 120° of the hip. The second one is the vertical position, with an angular range of 120° to 150°. The third and last position is called extended position, in which angulation ranges between 150° and 180°.

The most important muscles to perform the technical maneuver of dinghy hiking are the quadriceps; the abdominals (transverse, anterior rectus, major oblique, and minor oblique); and, to a lesser extent, the sternocleiodomastoid, iliopses, and anterior tibial [6,7,8,9]. The muscular action when performing this movement is called “quasi-isometric”, since there is no constant isometric action, as the sailor performs small modifications to adapt to the constant movements of the boat that are produced by the action of the wind and waves [10].

The use of simulators for the evaluation of strength, endurance [11], and performance [12] is widespread in sailing sports, and they allow reproducing the technical action of dinghy hiking.

Although the literature shows the relationship between performance and anthropometric characteristics, strength, and resistance in the hiking action, to our knowledge, no study has provided an equation to predict the performance of the sailor [5,7,10,13,14,15,16].

Therefore, the objective of our study was to analyse the relationship between performance and anthropometric, strength, and resistance measures in the action of dinghy hiking in a simulator and to provide an equation that can predict the position that the sailor will apply in the competition.

## 2. Materials and Methods

### 2.1. Participants

Twenty-nine Spanish male sailors of the Laser class voluntarily participated in our study, with an age range of 16 to 23 years. Three weeks prior to the day of the competition, all clubs, coaches, and sailors who were going to participate in the regatta were contacted via email. In that email, they received a document with the different tests they would perform in the study as well as the consent forms. Once they agreed to participate in the study, the author contacted the interested trainers and sailors again to set a date to perform the different tests and anthropometric measurements.

### 2.2. Ethical Approval

The characteristics and objectives of the study were explained to all the participants, both verbally and in writing, and informed consents were obtained from the participants and from the parents of the minor participants as well. The study protocol was approved by the Cádiz University Ethics Committee and met the requirements of the Declaration of Helsinki (1964) and the ethical standards in sports and exercise research.

### 2.3. Measurements and Procedures

The day before the competition, at the headquarters of the Andalusian Sailing Federation, the strength and resistance tests were performed in the simulator and the anthropometric measurements were recorded. Three stations were established, and groups of three athletes were formed, who performed each station, maintaining the group that had been established. The proceeding order was anthropometry, strength test, and resistance test. The last two were performed in a Laser class simulator. During data collection, they were also asked about their sailing experience, indicating the number of races in which they had participated in the previous year.

A Laser boat was used as the simulator on a transport car which had the wheels removed to give stability and to keep the structure fixed to the ground. The boat had a length of 4.23 m and a beam of 1.42 m, and the hull was made of fibreglass. The simulator was kept at a height of 75 cm measured from the board to the ground. In order for the sailors and evaluators to have a reference of the angulation that they should maintain, a vertical pole was placed with two horizontal extensions that marked the angulation between 150° and 180°.

For the anthropometric data collection, we used a SECA 703 scale and height gauge (SECA LTD., Hamburg, Germany), a measuring tape, and a pyrometer (Holtain Ltd., Crymych, UK). To calculate the percentage of body fat, we measured the triceps fold (mm) and subscapular fold (mm) [17]. The formula used to calculate the Body Mass Index (BMI) was weight/height^2^.

### 2.4. Anthropometry

An anthropometric assessment was performed following the indications established by the International Working Group of Kinanthropometry (IWGK) and currently collected by the International Society for the Advancement of Kinantropometry (ISAK) [18]. The measurements made were weight (kg), length (trochanteric, thigh, tibial, size, and sitting size), skin folds (triceps and subscapular), and perimeters (abdominal and upper thigh).

### 2.5. Simulator Strength Test

The athletes performed a progressive test of maximum load in the hiking action, which was considered valid if the subject managed to maintain the position for 5 s. The time that the participants had to hold the load was based on studies in which it is stated that the sailors can keep a maximum of twenty seconds in that position [5,19,20]. In addition, these authors indicate that modifications are made to the inclination of the trunk depending on the wind and swell conditions. Therefore, due to the fact that our test implied a progressive increase in the load, it was determined that the maximum time they could maintain that position would be 25% of that time in order to avoid any risk of injury to the athlete. For realization of the tests, the athletes were organized in groups of three. First, they performed a warm-up consisting of a 3-series circuit of 4 exercises and a 2-min break between series. The exercises were trunk flexions with elevated feet (15 repetitions), prone bridge with four supports (hold position 30 s), and obliques with scissors (15 repetitions). The hands were placed behind the head in the occipital area, and the trunk remained tilted in the extended position with an angulation between 150° and 180° with respect to the hip [5]. In the first execution, the participants did not carry any additional weight, while in the following repetitions, they carried a weight disk behind the head in the occipital area, which they held with their hands. The load progression was 2 kg in each attempt and 1 kg per attempt after reaching the threshold of 10 kg. Occasionally, due to the progression of the athlete, the load increase was reduced to 1 kg per attempt. Each athlete rested for 3 min until the next attempt. The test ended when the athlete could not hold the position for 5 s, and the last weight passed was taken as valid.

### 2.6. Simulator Resistance Test

Once the previous test was completed, each subject rested for 5 min before beginning the resistance test. The athlete was placed in the hiking position, with their hands behind their head, and the time that they could maintain the position with the same angulation between the hip and the trunk as in the strength test was recorded. Studies have shown that the hiking action requires a considerable isometric activity of the quadriceps and abdominal muscles for long periods of time and with little time to rest, which is the main cause of muscular fatigue [13]. The capacity to maintain this position is related to the performance of the sailor [14]. Therefore, the objective of this test was to evaluate the resistance of the musculature involved in the hiking action.

### 2.7. Statistical Analyses

The data are presented as means and standard deviations (SD). The level of significance was set at *p* ≤ 0.05. SPSS 20.0 (SPSS Lead Technologies Inc., Chicago, IL, USA) was used for the statistical analyses. The data were subjected to a descriptive analysis and inferences; their normality was also verified using the Kolmogorov–Smirnov test. For the variables that did not follow a normal distribution, the Mann–Whitney U nonparametric test was performed. A multiple linear regression analysis was carried out with the aim of examining the association between spatial orientation and performance, considering the following as predictive variables: age, navigation experience, strength test, resistance test, body weight, size, sitting height, Body Mass Index (BMI), body fat percentage, trochanteric length, thigh length, tibial length, foot length, abdominal perimeter, and upper thigh perimeter. The forward stepwise regression method was used to perform the multiple linear regression analysis. The performance prediction value of the obtained equations was verified using the Student’s T test between the real ranking variables and in the predicted ranking with this equation. Next, based on the results obtained in the strength and resistance tests, four groups were determined: high-strength level, low-strength level, high-resistance level, and low-resistance level. The group with a high-strength level was the one composed of those who bore a load equal to or greater than 9 kg, and the group with a low-strength level was the one that presented values below 9 kg. The group with a high-resistance level maintained the hiking position for 60 s or longer, while the group with a low-resistance level remained in that position for less than 60 s. To perform this division, we calculated the median value of the results obtained in the strength and resistance tests. The possible differences between the groups formed within each category (strength test and resistance test) were analysed. With the groups of high-strength level and high-resistance level, the multiple linear regression analysis was performed again including the same predictor variables; then, these groups were compared to see if there were differences between them.

## 3. Results

Table 1 shows the descriptive analysis of the variables analysed for the total sample and for each of the groups. It is observed that the sailors of the group with a higher strength level had greater height, a better result in the endurance test, and greater leg length compared to the group of low-strength level. Between the groups of high-resistance level and low-resistance level, there were differences only in the strength test, with those of the first group being the ones who had better results. When comparing the sailors of the high-strength group with those of the high-resistance group, no differences were observed in any of the analysed variables.

Table 2 shows the results of the multiple linear regression analysis using the forward stepwise method for the total sample. We obtained a two-step model to find the optimal model. The model obtained showed a linear relationship of 56% and a goodness of fit of R^2^ = 0.53. This model includes the constant, the age, and sailing experience, excluding the rest of the variables. The equation obtained with this model to predict the performance is as follows: performance = 61.398 + (−2.911 × age) + (−4.680 × sailing experience). When checking the prediction value of the performance obtained with this equation, using the Student’s T test to compare the relationship between the real ranking and the ranking estimated by this equation, we observed that the margin of error was only −0.001.

In the multiple linear regression analysis for the high-strength group using the forward stepwise method, the optimal model is shown in three steps. This model has a linear relationship of 81% and a goodness of fit of R^2^ = 0.76. The variables included in the last model are the constant, height, age, and thigh length (Table 3). The equation for this model is as follows: performance = 311.971 + (−1.089 × height) + (−1.946 × age) + (−1.537 × thigh length). The margin of error was also very low when comparing the relationship between the real and calculated rankings (−0.03), whereas the correlation obtained between the two rankings was very high (r = 0.9; *p* < 0.01).

The multiple linear regression analysis for the high-resistance group presents a two-step model, with a linear relationship of 56% and a corrected R^2^ = 0.5 (Table 4). The variables included in the model obtained in step two are constant, age, and sailing experience. The equation for this model is as follows: performance = 83.994 + (−2.193 × age) + (−7.209 × sailing experience). In this case, the margin of error obtained was −0.005.

## 4. Discussion

The aim of this study was to investigate how performance could be influenced by the sailor’s anthropometric characteristics, strength, and endurance when performing the technical maneuver of dinghy hiking as well as to determine an equation to predict such performance.

The sailors of the group of high-strength level had greater height and greater thigh length compared to those of the group of low-strength level (Table 1). These differences in anthropometric characteristics seem to mark a beneficial difference when it comes to dinghy hiking. Sailors with greater trunk height and length will perform the hiking action more efficiently [7,21]. This efficiency of the maneuver could justify the fact that the high-level group also presented a better result in the resistance test. It must also be taken into account that the subjects with more strength in the muscles involved in maintaining this position will tend to maintain the hiking position for a longer time, since their trunk weight will imply a lower relative load. These authors also indicate that having greater body weight will be equally positive, although in our case, we found no differences between the two groups in that variable.

In the analysis between the high- and low-resistance groups, the high-level group showed greater strength in the musculature involved in the hiking action (Table 1). This could be explained in the same way as it was done between the high- and low-strength groups. In this regard, sailors specifically develop the musculature involved in the hiking maneuver by being trained in this action [22], and expert sailors have greater resistance than amateur sailors when performing this action [14]. The study of the musculature involved in hiking shows how elite sailors are more efficient in stabilizing muscle demand and in delaying the onset of fatigue after carrying out this action for twenty minutes [15]. These conclusions have been drawn by other authors, who also indicate that fatigue follows a linear trend until exhaustion, thus relating isometric muscle action to the neuromuscular fatigue rate [16]. This importance of the variable strength in the hiking action for the Laser class could be an indicator of a more dynamic muscular activity instead of a static activity, thus being in opposition to isometry [13]. Having high-strength levels in the musculature involved in hiking may reduce the relative isometric load in this action, allowing the sailor to tolerate and maintain isometric activity in the most optimal conditions. Another aspect that benefits resistance when hiking is maximum strength in the abdominal muscles, fulfilling two important functions. On the one hand, there would be the stabilization of the spine and, on the other hand, the optimization of trunk flexion and extension [23]. Therefore, those sailors who have higher levels of strength in this musculature may have a greater resistance potential to maintain for a longer time the inclined position of the trunk outside the boat’s side.

The fact that no differences were found when comparing the groups of high-strength level and the group of high-resistance level was logical, since both were high-level groups. They can be considered to have some homogeneity in the analysed variables.

The results of the regression analysis in the total sample show that the variables age and sailing experience are the ones that will determine the performance with a positive relationship. Therefore, as the sailor’s age and experience increase, so does their chances of obtaining a better competitive result. Regarding the most significant variable in the regression analysis, the age of the sailor seems to be more decisive than their experience (−0.653 vs. −0.422). With the regression equation obtained through this model, it is possible to explain 53% of the performance of the sailor in the Laser class = 61.398 + (−2.911 × age) + (−4.680 × sailing experience). It should be noted that this is the first study to provide an equation that allows predicting the position that the sailor will implement during the competition.

Some studies have described the relationship between performance and age, also with elite male sailors [24]. Age seems to be a key factor in choosing the most suitable route during the competition. The age of the sailor is significantly and negatively related to the distance travelled (r = −0.88; *p* < 0.01), with the oldest sailors being those who will use less time and distance to complete the competition course [25,26].

With respect to the experience of the sailor, based on weekly training hours, it could allow a better handling of the boat to implement tactical procedures and to have a better knowledge of the climatic conditions. Another benefit for those who have more experience is at the time of departure and in the orientation toward the windward buoy. The most experienced sailors will have greater fixation in relevant points during departures, being able to perform it more quickly and to avoid crossing with their rivals, a fact that could hinder arrival to the windward buoy. From a psychological point of view, age and navigation experience are related to performance based on navigation tactics [27]. Older sailors will have spent more time practicing this sport, either in training or in competition; thus, they will have gained more experience. This experience will allow them to develop greater confidence about having the necessary resources to respond to the stimuli during training and competition.

In our multiple linear regression results for the group of high-strength levels and high-resistance levels, it is also observed that the age variable is included within the performance predictive equation. Age is the most powerful variable in both groups, having a Beta value = −0.532 (Table 3) in the high-strength group and −0.550 (Table 4) in the high-resistance group.

In the high-resistance group, age and navigation experience are in the multiple linear regression model. As in the regression analysis with the total sample, with these two variables, it is possible to explain 56% of the performance, and the equation obtained to predict the performance is 83.994 + (−2.193 × age) + (−7.209 × sailing experience).

Based on all of the above and observing the multiple linear regression analysis performed for each group, we can assert that age and sailing experience are key variables for the performance of the sailor in the Laser class.

The multiple linear regression analysis in the high-strength group also gives us two variables that could be related to the ability of the sailor to carry out the hiking action. In addition to age, this analysis includes height and thigh length as predictive variables of performance (Table 3). The equation obtained is 311.971 + (−1.089 × height) + (−1.946 × age) + (−1.537 × thigh length). This equation explains 76% of the performance of the sailor, thus being the best of the three equations obtained in this study.

The height of the sailor and the length of the thigh will allow the sailor to move more its center of mass, in addition to having a greater lever arm when doing hiking bench, thus producing a greater moment of forces to counteract the heel of the boat. The key in this interaction of forces is the moment produced by the sailor when making a hiking maneuver and the horizontal distance that exists between the center of mass of the sailor and the center of the boat [21]. The greater the distance, the greater the lever arm that the sailor will exercise. Although we found a relationship between performance and height, we did not find a relationship between performance and sitting height [7,28]. Therefore, the height of the sailor and the length of the thigh will influence the moment generated when hiking, since they will condition the position of the sailor’s center of mass and the distance to the center of the boat.

Although our first results have not shown any relationship with performance, if we divide the sailors into two groups based on their competitive result and compare them, it is observed that those who belong to the group with the highest performance have greater age, weight, and sitting height (results not shown). The age variable is also decisive to differentiate between sailors of higher and lower levels, with the emergence of two new variables related to body composition: weight and sitting height. Like height, the weight and sitting height benefit the sailor in their ability to produce more momentum with their lever arm. A study with sailors of the Finn class showed that the highest level sailors have a greater body mass compared to those of lower levels, basing their level on the results obtained in competition [29]. The maximum strength at performing the hiking maneuver has also been studied previously in a simulator with elite sailors of the Laser class, and their results showed a significant negative relationship between performance and the body weight of the sailor, where sailors with greater body weight showed better competitive results [13]. The hiking action is key to the performance of the sailor, and the height of the subject together with the sitting height will determine the moment of the pair of forces involved during the action. The moment of force produced by the sailor can be calculated by measuring the product between the weight of the sailor and the horizontal distance between the center of gravity of the subject and the center of the boat [9]. Therefore, the higher the height and the sitting height, the greater the moment produced when dinghy hiking. Similarly, having a greater weight will also help to produce a greater moment of strength. When performing the hiking action, the distance from the center of gravity and the moment produced are considered more important than the angulation of the hip and knee joints [7]. Based on previous studies, it is determined that the optimal height of the sailor ranges between 166 and 188 cm, while the weight should be around 65–84 kg [2,4,6,11,14,20,30]. It must be taken into account that the height of the sailor can become a negative factor when it exceeds the relationship between the weight of the boat and the weight of the sailor [31].

One of the main limitations of our study was the sample size. It would have been interesting to have a larger sample of sailors and to have a representative sample of the female gender. Another limitation was the fact that we did not have a validated test to evaluate the variables of strength and endurance in the muscles involved in the hiking action. Furthermore, other functional tests could have helped to strengthen the results obtained in the study.

## 5. Conclusions

The variables that are most related to performance are age and sailing experience. Regarding the variables related to body composition, the results show that the sailor’s height and thigh length are key to their performance. With the equation obtained from the multiple linear regression analysis in the group of sailors of high-strength level, it is possible to explain 76% of the performance. Such an equation is as follows: performance = 311.971 + (−1.089 × height) + (−1946 × age) + (−1.537 × thigh length).

## Figures and Tables

**Table 1 ijerph-16-04920-t001:** Mean ± SD of the variables analysed in all sailors and in the groups of high- and low-strength levels and high- and low-resistance levels.

Variable	Total Sample (*n* = 29)	High-Strength Level (*n* = 15)	Low-Strength Level (*n* = 14)	High-Resistance Level (*n* = 15)	High-Resistance Level (*n* = 14)
Age (years)	17 ± 3	17.6 ± 3.4	16.3 ± 2.5	16.8 ± 3.2	17.1 ± 2.9
Sailing experience	4 ± 1.2	4.3 ± 1.1	3.7 ± 1.2	4.4 ± 0.8	3.5 ± 1.4
Height (cm)	172.4 ± 6.4	175.1 ± 5.7 **	168.5 ± 4.9	172.9 ± 7.6	171.8 ± 5.1
Sitting height (cm)	88.2 ± 3.8	89.4 ± 3.8	86.9 ± 3.5	87.8 ± 4.6	88.7 ± 2.9
Strength test (kg)	9.8 ± 5.3	13.6 ± 4.6	5.7 ± 2.1	12.6 ± 5.4 ^†^	6.8 ± 3.2
Resistance test (s)	52.1 ± 23.1	64.1 ± 17.7 **	39.3 ± 21.7	70.3 ± 11.1	32.6 ± 15
Weight (kg)	66.4 ± 10.1	68.3 ± 8.9	64.3 ± 11.1	64.2 ± 9.4	68.7 ± 10.5
BMI (kg/m^2^)	22.3 ± 3.3	22.1 ± 3	22.6 ± 3.6	21.4 ± 2.7	23.3 ± 3.7
% body fat	23.2 ± 12.1	20.7 ± 9.6	25.8 ± 15	20.4 ± 8,3	26.1 ± 15.7
Leg length (cm)	92.6 ± 6	95.1 ± 4.9 **	89.9 ± 6.1	92.6 ± 6.6	92.6 ± 5.5
Thigh length (cm)	46.8 ± 3.8	47.8 ± 3	45.6 ± 4.3	47.5 ± 3.1	46 ± 4.3
Tibia length (cm)	39.5 ± 2.7	40.2 ± 2.3	38.7 ± 3.1	39.6 ± 2.6	39.4 ± 3
Foot length (cm)	26.6 ± 2	27.1 ± 1.8	26.1 ± 2.1	26.6 ± 1.8	26.7 ± 2.3
Abdominal perimeter (cm)	78.6 ± 11.8	77.8 ± 13.4	79.4 ± 10.2	75.5 ± 12.3	81.9 ± 10.7
Upper thigh perimeter (cm)	54.4 ± 5.1	54.7 ± 4.3	54.1 ± 5.9	53.9 ± 4.8	55 ± 5.4

Note: ** *p* < 0.01 for the comparison between high- and low-strength groups; ^†^
*p* < 0.01 for the comparison between high- and low-resistance groups.

**Table 2 ijerph-16-04920-t002:** Coefficients of the multiple linear regression model using the forward stepwise method for the total sample of sailors.

Model		Non-Standardised Coefficients		Typified Coefficients	t	Sig.	Confidence Interval 95% for B	
		B	Std. Error	Beta			Lower Bound	Upper Bound
2	(Constant)	61.398	11.603		6.918	0.000	58.186	107.381
	Age	−2.911	0.580	−0.653	−5.018	0.000	−4.104	−1.719
	Sailing experience	−4.680	1.445	−0.422	−3.239	0.003	−7.649	−1.710

Method used: forward stepwise. B = linear regression coefficient; Std. Error = estimated error; t = statistical significance; Beta = standardised partial regression coefficient; Lower Bound = lower limit; Upper Bound = upper limit; Sig = level of significance.

**Table 3 ijerph-16-04920-t003:** Coefficients of the multiple linear regression model using the forward stepwise method in the high-strength group.

Model		Non-Standardised Coefficients		Typified Coefficients	t	Sig.	Confidence Interval 95% for B	
		B	Std. Error	Beta			Lower Bound	Upper Bound
3	(Constant)	311.971	54.507		5.724	0.000	192.003	431.939
	Height	−1.089	0.303	−0.498	−3.600	0.004	−1.755	−0.423
	Age	−1.946	0.508	−0.532	−3.834	0.003	−3.063	−0.829
	Thigh length	−1.537	0.550	−0.373	−2.795	0.017	−2.747	−0.327

Method used: forward stepwise. B = linear regression coefficient; Std. Error = estimated error; t = statistical significance; Beta = standardised partial regression coefficient; Lower Bound = lower limit; Upper Bound = upper limit; Sig = level of significance.

**Table 4 ijerph-16-04920-t004:** Coefficients of the multiple linear regression model using the forward stepwise method in the high-resistance group.

Model		Non-Standardised Coefficients		Typified Coefficients	t	Sig.	Confidence Interval 95% for B	
		B	Std. Error	Beta			Lower Bound	Upper Bound
2	(Constant)	83.994	17.769		4.727	0.000	45.278	122.709
	Age	−2.193	0.762	−0.550	−2.877	0.014	−3.854	−0.532
	Sailing experience	−7.209	2.930	−0.470	−2.461	0.030	−13.592	−0.826

Method used: forward stepwise. B = linear regression coefficient; Std. Error = estimated error; t = statistical significance; Beta = standardised partial regression coefficient; Lower Bound = lower limit; Upper Bound = upper limit; Sig = level of significance.

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
