# Peer review of "Performance Factors in Dinghy Sailing: Laser Class"

_ijerph, 2019, doi:10.3390/ijerph16244920_

Round 1
Reviewer 1 Report
Thank you for sharing your research results. This is very interesting material, but some corrections and clarifications should be made at work. Below I will raise my doubts.
Statistical analysis:
- page 3 line 122 - correct sentence: "The level of significance was "set at" p <= 0.05"
- why was the Kruskal-Wallis test used to compare two independent groups? The best equivalent is the Mann-Whitney U test;
- I understand from the above that the purpose of applying analysis of variance was to compare 4 groups; but what is the point of comparing 4 groups that contain the same objects ???
If you have more than one independent variables it would be multiple regression analysis - not simply linear.
Please read about:
In statistics, linear regression is a linear approach to modeling the relationship between a scalar response (or dependent variable) and one or more explanatory variables (or independent variables). The case of one explanatory variable is called simple linear regression. For more than one explanatory variable, the process is called multiple linear regression. [1] This term is distinct from multivariate linear regression, where multiple correlated dependent variables are predicted, rather than a single scalar variable. [2]
David A. Freedman (2009). Statistical Models: Theory and Practice. Cambridge University Press. p. 26. "A simple regression equation has on the right hand side an intercept and an explanatory variable with a slope coefficient. A multiple regression equation has two or more explanatory variables on the right hand side, each with its own slope coefficient"
Rencher, Alvin C. .; Christensen, William F. (2012), "Chapter 10, Multivariate regression - Section 10.1, Introduction", Methods of Multivariate Analysis, Wiley Series in Probability and Statistics, 709 (3rd ed.), John Wiley & Sons, p. 19, ISBN 9781118391679.
- specify "Stepwise" method; Was it a forward or backward regression?
- on what basis were the division of sailors into groups? Why was 9 kg and 60 seconds taken as the criterion for division?
In table 1 specify what the symbols used in the footer apply to.
Table 2-3-4 - there is no need to present the regression model obtained in each step, the final best model should be presented.
Correct the descriptions under Tables 2 and 3 and 4
- line 164 "t = t de Student" - I don't know what's going on?
- line 165 "Lower Bown" replace with "Lower Bound"
In addition, I attach a file with the markings to which you should refer.
Author Response
Dear reviewer,
We want to thank the editorial team for giving us the opportunity to review and improve our manuscript, and the editors and reviewers for their thoughtful and constructive comments. We have considered all suggestions and incorporated them into the revised manuscript. The changes in the original manuscript are highlighted by controlling changes in the word processor, and we believe that our manuscript is stronger as a result of these modifications. Below are the responses detailing the comments of the reviewers point by point.
REVIEWER 1
Point 1: Thank you for sharing your research results. This is very interesting material, but some corrections and clarifications should be made at work. Below I will raise my doubts.
Statistical analysis:
- page 3 line 122 - correct sentence: "The level of significance was "set at" p <= 0.05"
Response 1: Modified
Point 2: why was the Kruskal-Wallis test used to compare two independent groups? The best equivalent is the Mann-Whitney U test;
Response 2: The variables have been evaluated according to the Mann-Withney test.
Point 3: I understand from the above that the purpose of applying analysis of variance was to compare 4 groups; but what is the point of comparing 4 groups that contain the same objects ???
Response 3: The objective of this analysis was to compare whether there was a variable that clearly differentiated each of the groups based on strength and endurance. For example, we found it interesting to discover that the high strength group also had better results in the resistance test compared to the low strength group.
Point 4: "... A linear regression analysis was carried out with the objective of examining the association between spatial orientation and performance, considering as predictive variables age, navigation experience ...".
If you have more than one independent variables it would be multiple regression analysis - not simply linear.
Response 4: We performed a multiple regression analysis, but we made the mistake of not indicating it in the text. The word “multiple” has been included.
Point 5: Please read about:
In statistics, linear regression is a linear approach to modeling the relationship between a scalar response (or dependent variable) and one or more explanatory variables (or independent variables). The case of one explanatory variable is called simple linear regression. For more than one explanatory variable, the process is called multiple linear regression. [1] This term is distinct from multivariate linear regression, where multiple correlated dependent variables are predicted, rather than a single scalar variable. [2]
David A. Freedman (2009). Statistical Models: Theory and Practice. Cambridge University Press. p. 26. "A simple regression equation has on the right hand side an intercept and an explanatory variable with a slope coefficient. A multiple regression equation has two or more explanatory variables on the right hand side, each with its own slope coefficient"
Rencher, Alvin C. .; Christensen, William F. (2012), "Chapter 10, Multivariate regression - Section 10.1, Introduction", Methods of Multivariate Analysis, Wiley Series in Probability and Statistics, 709 (3rd ed.), John Wiley & Sons, p. 19, ISBN 9781118391679.
Response 5: The procedure has been correct but we have not indicated the correct concept. It has been modified and indicated that we have performed a multiple regression analysis. We have included the term “multiple”. Thank you for the bibliography provided on statistics, it has been a great help.
Point 6: specify "Stepwise" method; Was it a forward or backward regression?
Response 6: We have used the “stepwise forward” method. Modified.
Point 7: on what basis were the division of sailors into groups? Why was 9 kg and 60 seconds taken as the criterion for division?
Response 7: To perform this division, the median value was calculated in the results obtained in the simulator strength and resistance test.
Point 8: In table 1 specify what the symbols used in the footer apply to.
Response 8: Modified.
Point 9: Table 2-3-4 - there is no need to present the regression model obtained in each step, the final best model should be presented.
Response 9: Modified.
Point 10: Correct the descriptions under Tables 2 and 3 and 4
- line 164 "t = t de Student" - I don't know what's going on?
Response 10: It´s a mistake. It has been changed for statistical significance. Modified.
- line 165 "Lower Bown" replace with "Lower Bound"
Response 10: Modified.

Reviewer 2 Report
The objective of the study was to investigate how the performance can be influenced by the anthropometric characteristics of the athletes. Furthermore, the efficiency of performance in strength and endurance training could predict the ability to excel in sports performance.
The analysis of these characteristics of the athletes could represent the starting point for the formulation of an equation, based on linear regressions and statistical analysis of the data, which can generally allow the researchers to categorize the athletes in high or medium-low performance categories.
This could be an interesting approach. However, some sections of the manuscript should be improved.
Major revisions:
1) It seems quite obvious that the navigation experience and age (correlated with the sporting history) are two fundamental parameters for the good performance in this discipline.
However, more interesting would be a deep evaluation of the weight height and sitting position parameters in the performance. (pag 7 lane 277 - 283)
2) The authors must better explain the resistance test. Is this a validated test? What physiological parameters have been evaluated?
Minor revisions:
1) Other functional tests have been performed (i.e. Reaction time test)?
In that case, it should be entered.
2) There are some typing errors and some missing references in the text.
Author Response
Dear reviewer,
We want to thank the editorial team for giving us the opportunity to review and improve our manuscript, and the editors and reviewers for their thoughtful and constructive comments. We have considered all suggestions and incorporated them into the revised manuscript. The changes in the original manuscript are highlighted by controlling changes in the word processor, and we believe that our manuscript is stronger as a result of these modifications. Below are the responses detailing the comments of the reviewers point by point.
REVIEWER 2
The objective of the study was to investigate how the performance can be influenced by the anthropometric characteristics of the athletes. Furthermore, the efficiency of performance in strength and endurance training could predict the ability to excel in sports performance.
The analysis of these characteristics of the athletes could represent the starting point for the formulation of an equation, based on linear regressions and statistical analysis of the data, which can generally allow the researchers to categorize the athletes in high or medium-low performance categories.
This could be an interesting approach. However, some sections of the manuscript should be improved.
Major revisions:
Point 1: It seems quite obvious that the navigation experience and age (correlated with the sporting history) are two fundamental parameters for the good performance in this discipline.
However, more interesting would be a deep evaluation of the weight height and sitting position parameters in the performance. (pag 7 lane 277 - 283)
Response 1: We have discussed the influence of the variables weight, height and height sitting with the performance. (pag 8 lane 316-328).
Point 2: The authors must better explain the resistance test. Is this a validated test? What physiological parameters have been evaluated?
Response 2: The test is not validated, but other similar test have been taken as reference in which the resistance of the main muscles has been evaluated when hiking bench. We have not found any test validated with these characteristics for sailors and performed in a simulator. There is a test called “Bucket test” that is not validated. However, it has been used to evaluate the performance of the sailor based on the isometric action of the quadriceps muscles (Tan et al, 2008; Callewaert, et al, 2015). We decided not to use this test because it only evaluates the quadriceps muscles. (pag 3 lane 120-124).
Tan, B.; Aziz, A.R.; Spuway, N.C.; Toh, C.; Mackie, H.; Xie, W.; Wong, J.; Fuss, F.K.; Teh, K.C. Indicators of maximal hiking performance in Laser sailors. Eur. J. Appl. Physio, 2006, 98(2), 169-176. [Pubmed]
Callewaert, M; Boone, J.; Celie, B.; De Clercq, D.; Bourgois, J.G. Indicators of sailing performance in youth dinghy sailing. Eur. J. Sport Sci, 2015, 15(3), 213-219. [Pubmed]
Minor revisions:
Point 1: Other functional tests have been performed (i.e. Reaction time test)?
In that case, it should be entered.
Response 1: Other types of test have not been performed.
Point 2: There are some typing errors and some missing references in the text.
Response 2: The indicated errors have been reviewed.
Round 2
Reviewer 2 Report
I thank the authors for responding satisfactorily to all the critical points highlighted.
However, I think that in order to improve the quality of work it would have been better to use a validated resistance test and also some other functional tests like a reaction time test.
Author Response
Dear Reviewer,
Again I thank you for your guidance to improve our manuscript.
I enclose a certificate of the translation review by a professional.
The following sentence has been included in the abstract:
"Performance in the Laser class will be determined by the tactics (age and sailing experience) and the morphological characteristics of the sailor (height and sitting height)."
The limitations of the study have been described in the discussion and it has been specified that a validated test was not used and that it would have been interesting to perform other functional tests:
"One of the main limitations of our study was the sample size. It would have been interesting to have a larger sample of sailors and have a representative sample of the female gender. Another limitation was the fact that we did not have a validated test to evaluate the variables of strength and endurance in the muscles involved in the hiking action. Furthermore, other functional tests could have helped to strengthen the results obtained in the study."
